# Influence of the Cumulative Incidence of COVID-19 Cases on the Mental Health of the Spanish Out-of-Hospital Professionals

**DOI:** 10.3390/jcm11082227

**Published:** 2022-04-15

**Authors:** Raúl Soto-Cámara, Susana Navalpotro-Pascual, José Julio Jiménez-Alegre, Noemí García-Santa-Basilia, Henar Onrubia-Baticón, José M. Navalpotro-Pascual, Israel John Thuissard, Juan José Fernández-Domínguez, María Paz Matellán-Hernández, Elena Pastor-Benito, Carlos Eduardo Polo-Portes, Rosa M. Cárdaba-García

**Affiliations:** 1Department of Health Sciences, University of Burgos, 09001 Burgos, Spain; rscamara@ubu.es (R.S.-C.); ngarcia@saludcastillayleon.es (N.G.-S.-B.); monrubia@saludcastillayleon.es (H.O.-B.); mmatellanh@saludcastillayleon.es (M.P.M.-H.); 2Emergency Medical Service of Castilla y León—Sacyl, 47007 Valladolid, Spain; 3Red de Investigación de Emergencias Prehospitalarias (RINVEMER), Sociedad Española de Urgencias y Emergencias (SEMES), 28020 Madrid, Spain; juljimal@gmail.com (J.J.J.-A.); chemanp@gmail.com (J.M.N.-P.); israeljohn.thuissard@universidadeuropea.es (I.J.T.); jjfernandezd@gmail.com (J.J.F.-D.); elenapastorbenito@gmail.com (E.P.-B.); carlospolo1763@hotmail.com (C.E.P.-P.); 4Emergency Medical Service of Madrid—SUMMA 112, 28045 Madrid, Spain; 5Nursing Department, Faculty of Medicine, Autonomous University of Madrid, 28029 Madrid, Spain; 6Faculty of Medicine, Alfonso X El Sabio University, 28691 Madrid, Spain; 7Department of Medicine, Faculty of Biomedical and Health Sciences, European University of Madrid, 28670 Madrid, Spain; 8Emergency Service, HLA Moncloa University Hospital, 28008 Madrid, Spain; 9Nursing Department, Faculty of Nursing, University of Valladolid, 47005 Valladolid, Spain

**Keywords:** coronavirus infections, health personnel, emergency medical services, psychological stress, anxiety, depression, self-efficacy, incidence

## Abstract

This study aimed to analyze the psychological affectation of health professionals (HPs) of Spanish Emergency Medical Services (EMSs) according to the cumulative incidence (CI) of COVID-19 cases in the regions in which they worked. A cross-sectional descriptive study was designed, including all HPs working in any EMS of the Spanish geography between 1 February 2021 and 30 April 2021. Their level of stress, anxiety and depression (DASS-21) and the perception of self-efficacy (G-SES) were the study’s main results. A 2-factor analysis of covariance was used to determine if the CI regions of COVID-19 cases determined the psychological impact on each of the studied variables. A total of 1710 HPs were included. A third presented psychological impairment classified as severe. The interaction of CI regions with the studied variables did not influence their levels of stress, anxiety, depression or self-efficacy. Women, younger HPs or those with less EMS work experience, emergency medical technicians (EMT), workers who had to modify their working conditions or those who lived with minors or dependents suffered a greater impact from the COVID-19 pandemic in certain regions. These HPs have shown high levels of stress, anxiety, depression and medium levels of self-efficacy, with similar data in the different geographical areas. Psychological support is essential to mitigate their suffering and teach them to react to adverse events.

## 1. Introduction

The declaration of the disease caused by the virus SARS-CoV-2, named COVID-19, as “The Sixth International Public Health Emergency” and the proclamation of the resulting situation as a pandemic by the World Health Organization (WHO) has produced important changes at the economic, social and health levels in all countries [1,2]. As in other areas, in Spain, this affectation has not remained uniform over time but has fluctuated depending on the cumulative incidence (CI) of cases in the different waves and geographical regions. For this, the Center for the Coordination of Health Alerts and Emergencies of the Ministry of Health and the National Epidemiological Surveillance Network of the National Center for Epidemiology of the Carlos III Health Institute have been designated as the organizations in charge of collecting CI information in the different Spanish regions and monitoring possible change from the start of the COVID-19 pandemic to the present [3,4].

During this period, numerous studies have analyzed the impact of the COVID-19 pandemic on the mental health of the general population and certain sectors [5,6]. Health professionals (HPs) have been among the most affected groups, focusing most of this research on primary care or hospital workers and specific national care models [7,8,9,10,11]. HPs have faced very intense and stressful work situations, such as work overload, prolonged work shifts, fewer hours of rest, no clear and defined protocols for action, strict safety instructions and measures, the constant need for concentration and vigilance, the lack of personal protective equipment and reduced social contact, as well as having to perform tasks for which many professionals have not been prepared [12,13]. This situation of stress has put both the physical and mental health of the HPs at risk. Their general well-being has been altered, and they have started to show high levels of anxiety and depression, other emotional disorders, sleep problems, difficulty in interpersonal relations, dysfunctional cognitive reactions, substances use behaviors, post-traumatic stress, and even vicarious traumatization stemming from compassion towards the patients that they were treating [14,15,16,17].

Generally, emergency medical service (EMS) is the department in charge of out-of-hospital care for critically ill patients in most countries. During the COVID-19 pandemic, this service had had to develop new policies, procedures, and protocols to address the consequences of this epidemiological situation, characterized by an increase in the volume of calls and the care of patients with suspected signs or confirmed cases [18]. However, the specific scientific research referring to out-of-hospital EMSs has been very limited [18,19], even though they continue to be one of the frontline healthcare providers. Like HPs in other settings, the findings of these studies have shown a negative impact of the COVID-19 pandemic on the mental health of out-of-hospital workers, with an increase in the prevalence of disorders due to stress, anxiety, depression, insomnia or burnout [18,19].

For all these reasons, the objective of the present study was to analyze the level of psychological affectation of the HPs of the Spanish EMSs, according to the CI of COVID-19 cases of the geographical regions in which they worked.

## 2. Materials and Methods

### 2.1. Study Design-Participants

A cross-sectional descriptive study was designed. The study population was all HPs working in any EMS in the Spanish geography between 1 February and 30 April 2021. Not accepting voluntary participation in the study or not completing the entire questionnaire were considered exclusion criteria.

For the estimation of the sample size, it was considered that 23,467 HPs worked in EMS in Spain in 2020, according to data from the Statistical Portal of the Primary Care Information System of the Ministry of Health [20]. It was necessary to recruit at least 1066 subjects to achieve a confidence level of 95% and an accuracy of 3%, considering a 15% possible loss.

### 2.2. Procedure—Data Collection

Participants were selected using non-probabilistic snowball sampling. An online questionnaire was used for data collection. The link to the questionnaire was distributed through the Prehospital Emergency Research Network (RINVEMER) of the Spanish Society of Emergency Medicine (SEMES) and the managers of the different EMSs. In the first part of the questionnaire, the participants were informed of the characteristics and objectives of the study and its anonymous and voluntary nature. Its completed return implied the person’s informed consent to participate in the research. To guarantee the anonymity of the HPs, no personal data was collected that could allow their identification, even in those cases in which they specifically requested feedback on the results obtained, for which a personal alphanumeric code was created. Participants could withdraw from the study at any time without giving any reason. The time required to answer the questionnaire was approximately 15–20 min. All doubts were resolved by email.

The research protocol was approved by the Medicine Ethics and Research Committee of the East Valladolid Health Area (PI-20-2052), respecting the principles of the Declaration of Helsinki and its successive revisions [21].

### 2.3. Main Outcomes—Instruments

The study’s main results were the level of stress, anxiety and depression of the HPs and their perception of self-efficacy.

The reduced version of the Depression Anxiety Stress Scale (DASS-21) was used as a self-reported instrument to assess the intensity of 21 different symptoms associated with a negative emotional state [22]. It consists of 3 subscales, with 7 items each one: (i) stress, which evaluates tension, irritability, nervousness, impatience, agitation, and negative affect; (ii) anxiety, which assesses physiological activation, musculoskeletal symptoms and subjective sensation of anxiety; and (iii) depression, which evaluates hopelessness, dysphoria, sadness, anhedonia, low self-esteem, and low positive affect. The HPs must indicate the frequency with which they have experienced these symptoms in the previous 2 weeks using a 4-point Likert scale (0: Never; 3: Always). In each subscale, the total score is obtained by adding the points of each item and multiplying it by 2. The score of the subscales ranges between 0 and 42, so the higher the value, the greater the degree of symptomatology. Similarly, this score can be categorized as normal, mild, moderate, severe or extremely severe. Its adaptation and validation to the Spanish population were carried out by Bados et al., with acceptable psychometric properties [23]. It has been widely used to assess the psychological impact of the COVID-19 pandemic on the general population [6] and HPs [18] as it has good discriminant validity in screening for mental disorders [24].

To evaluate the person’s perception of their ability to adequately handle different stressful situations, the Spanish adaptation of the General Self-Efficacy Scale (G-SES) was used [25,26]. It is made up of 10 items, with 10 response options (1: Never; 10: Always). The score ranges between 10 and 100, associating higher values with greater perceived self-efficacy. It presents good psychometric properties, with predictive capacity on coping styles, and internal consistency of 0.87 [26].

Other variables were also collected through an ad hoc questionnaire: sex, age, living with minors or dependent persons, professional category, previous work experience in EMS, change in working conditions, previous diagnosis of COVID-19 or CI of COVID-19 cases. For analytical purposes and based on the CI per 100,000 inhabitants defined by the Health Authorities on 1 February 2021, the Spanish geography was divided into 3 areas: region with low CI if ≤4999 cases, region with medium CI if 5000–6999 cases, and region with high CI if ≥7000 cases [27] (Figure 1).

### 2.4. Statistical Analysis

Categorical variables were summarized as absolute frequencies and percentages, while quantitative ones were in terms of mean and standard deviation (SD). The compliance of the normality criteria of the quantitative variables was evaluated using the Kolmogorov–Smirnov test; in those cases in which they did not follow a normal distribution, the criteria proposed by Blanca et al. were considered [28]. To contrast the levels of stress, anxiety, depression and self-efficacy in regions with the same CI of COVID-19 cases or between the 3 regions considered, the χ^2^ test, the Student’s *t*-test for independent samples, the one-way analysis of variance or the Pearson’s correlation were calculated, depending on the nature of the variables. For multiple comparisons, post hoc tests were corrected by Bonferroni adjustment. In addition, to find out if the different regions were a determining factor in the psychological impact of each of the variables, a 2-factor analysis of covariance (study variables × region) was performed. Statistical significance was considered if *p* < 0.05. Statistical analysis was carried out with SPSS version 25.0 software (IBM-Inc, Chicago, IL, USA).

## 3. Results

The sample consisted of 1710 participants; 50.58% were women, with a mean age of 43.54 years (SD ± 9.94). The most represented professional category was emergency medical technicians (EMT) (*n* = 765), followed by doctors (*n* = 474) and nurses (*n* = 453), with a mean work experience in EMS of 15.22 years (SD ±9.15). In relation to the mental health of these HPs, 37.39% (*n* = 639), 39.36% (*n* = 673) and 30.46% (*n* = 521) presented levels of stress, anxiety and depression categorized as severe or extremely severe. The mean scores obtained in stress, anxiety, depression and self-efficacy were 20.61 (SD ± 11.08), 13.08 (SD ± 11.17), 15.74 (SD ± 11.11) and 70.78 (SD ± 15.75), respectively. The distribution of their descriptive characteristics in the different CI regions is summarized in Table 1.

In areas with medium or high CI, women presented greater stress, anxiety, and depression; men who worked in areas with low CI reported less stress than those employed in areas with a higher number of COVID-19 cases. Regarding self-efficacy, men perceived higher values in areas with low CI. The interaction of gender and region did not affect the psychological variables analyzed (Table 2).

EMTs who worked in regions with low or high CI reported negative emotional states compatible with stress or depression more frequently than other professional categories. Their anxiety levels were also significantly higher in the three areas, regardless of the number of COVID-19 cases. The professional category and region combination did not influence the mean scores obtained on the DASS-21 and the G-SES (Table 3).

HPs who were forced to change their work schedule, location, or dedication reported higher levels of stress, anxiety and depression in the three regions of CI. When the psychological impact of the COVID-19 pandemic was analyzed, considering the need or not for changes in working conditions, it was concluded that the different regions were not a determining factor (Table 4).

Having a previous diagnosis of COVID-19 or living with minors and/or dependents was not related to significant changes in the values of stress, anxiety, depression and self-efficacy, except for a higher level of anxiety among those HPs with vulnerable dependents in regions with low CI. In both cases, when the interaction of these variables with the CI region was analyzed, no influence was observed on the psychological parameters (Table 5 and Table 6).

Both HPs’ age and EMS work experience were indirectly and weakly correlated with stress levels, anxiety, depression and self-efficacy, regardless of the number of COVID-19 cases in the region they worked (Table 7).

## 4. Discussion

This study is proposed to identify the impact of the COVID-19 pandemic on the mental health of HPs in Spanish EMSs and its influence on certain socio-demographic and labor variables, according to the number of cases registered in each region. The interaction of the CI regions with the other study variables considered has not altered the levels of stress, anxiety, depression and self-efficacy of the HPs, unlike what was observed by Brillon et al. [29]. However, in certain regions, a greater impact of the COVID-19 pandemic was observed on women, younger HPs or those with less EMS work experience, EMTs, workers who had to modify their working conditions or those who lived with minors or dependents.

Around a third of the participants presented severe or extremely severe levels of stress, anxiety and depression, data higher than those reported by HPs from other care settings [9,30,31,32]. Working on the front line, in areas where the unpredictability of the attended cases is greater or in environments with a high probability of contagion, as is the case of the EMSs, is becoming a risk factor for the development of negative responses to challenging situations [33,34]. The high number of HPs with scores in psychopathological alarm ranges should be considered a warning sign of the future psychosocial consequences of the acute phase of the COVID-19 pandemic, such as post-traumatic stress or burnout [35,36].

In this study, the levels of stress, anxiety or depression of the HPs have not been influenced by the number of COVID-19 cases declared in the different geographical areas. However, several authors have shown the existence of an “epicentric effect,” which explains a higher prevalence of these psychological conditions the closer the HPs are to the most affected regions [29,34,37,38,39,40]. Continuous exposure to stressful elements for long periods of time, lack of social support, or living the same reality as the patients have contributed to exacerbating this effect among HPs from regions with high CI [29].

A higher level of stress and emotional burden has been observed in women who worked in regions with medium or high CI and lower use of coping strategies. The less time dedicated to self-care or self-compassion and the high work pressure during the health emergency has led to the appearance and maintenance of this situation [41,42]. All this has been favored by factors such as gender discrimination, the progressive feminization of the health sector, the difficulties in reconciling work and family, the traditional assumption of the role of primary caregiver at home, the lack of sufficient support systems, the greater empathy in providing care, or the greater ability to express feelings to others and develop emotional responses to stressful events [41,43,44,45,46,47]. The lower psychological affectation of men, especially in regions with low CI, is related to their relative underreporting of symptoms and underuse of health services [48,49] and the widespread use of coping strategies focused on the problem. These strategies limit the ability to recognize their emotional difficulties and become aware of their own experiences [50].

Younger HPs and those with less EMS work experience were more vulnerable to developing symptoms compatible with stress, anxiety, or depression disorders, regardless of geographic area. Some authors have speculated that these workers, whatever their professional role, have less self-confidence and less resistance at a psychological level, and a greater degree of uncertainty in how to act in unforeseen and/or complex situations [37,41,51,52].

The psychological well-being of all participants has been affected during the COVID-19 pandemic. This finding suggests their great personal and emotional involvement, being more notable in the EMTs who worked in areas with low and high CI. The lesser affectation of doctors and nurses could be related to the use of coping strategies based on intellectualization and denial and greater resistance to somatization, related to personal achievements, professional experience or self-awareness [53,54].

The modification of working conditions increased the vulnerability of HPs to stress, anxiety and depression in all regions. The reorganization and restructuring of the EMSs and the adaptation of the workplace for health reasons may be the main causes of these changes. The sudden outbreak of COVID-19 has caused unpredictable changes in the work of HPs, with an increase in the demand for care, greater contact with patients suffering from serious and complex diseases, a reduction in rest times and a lack of socio-occupational support [29,55,56]. To deal with this situation, the EMSs have created units specifically dedicated to the care of patients with COVID-19 and have moved part of the HPs from one job to another [29,52]. The adaptation of these displaced HPs to this new context, in constant change, has caused them an additional mental burden [52]. On the other hand, some authors have observed a greater impact of the COVID-19 pandemic on HPs whose workplace had to be adapted. Among the possible explanations for this result are fear and lack of information about the interaction of their previous diseases with SARS-CoV-2 infection and its possible consequences in the medium and long term [57,58,59,60]. Furthermore, HPs with previous mental diseases are more likely to present this type of psychological symptoms [61].

Only HPs who lived with vulnerable people in regions with low CI had higher anxiety levels. This finding can be attributed to the fear of becoming infected and the consequent risk of transmitting the disease to their relatives [9,31,62].

The psychological discomfort of HPs must be considered beyond a merely individual level, as it directly impacts patient care. The most distressed HPs participate less in the therapeutic relationship, make more mistakes and even compromise clinical results [40]. Based on this premise, the need for health authorities to design psychological support strategies in which HPs reflect on their psycho-emotional reactions to adverse events is reinforced [63].

These results must be interpreted within the context of their limitations. It has not been possible to determine a causal relationship between variables due to the study’s cross-sectional nature. Psychological distress has been assessed only through self-report measures administered online, limiting access to HPs less accustomed to the use of new technologies. The use of non-probabilistic snowball sampling may have induced a self-selection bias by favoring the participation of HPs who are particularly sensitive to the issue and those who have a greater degree of affectation. Data collection lasted 12 weeks. This fact may have affected the quality of the responses since the CI of COVID-19 cases, and the perception related to the infection have differed between the first and last day. The lack of studies on this topic in the out-of-hospital setting has hindered comparing and contrasting the results obtained. Among its strengths is the collection of data from a large sample of HPs from all EMSs of the Spanish geography and the use of validated questionnaires with excellent psychometric properties.

## 5. Conclusions

The HPs from the Spanish EMSs present high levels of stress, anxiety, depression and medium levels of self-efficacy. Similar data were observed in different geographical areas. A greater impact of the COVID-19 pandemic has been observed on women, younger HPs or those with less EMS work experience, EMTs, workers who had to modify their working conditions or those who lived with minors or dependents in certain regions. In these HPs, psychological support is essential to mitigate their suffering, helping them to reflect on their psycho-emotional reactions to adverse events.

## Figures and Tables

**Figure 1 jcm-11-02227-f001:**
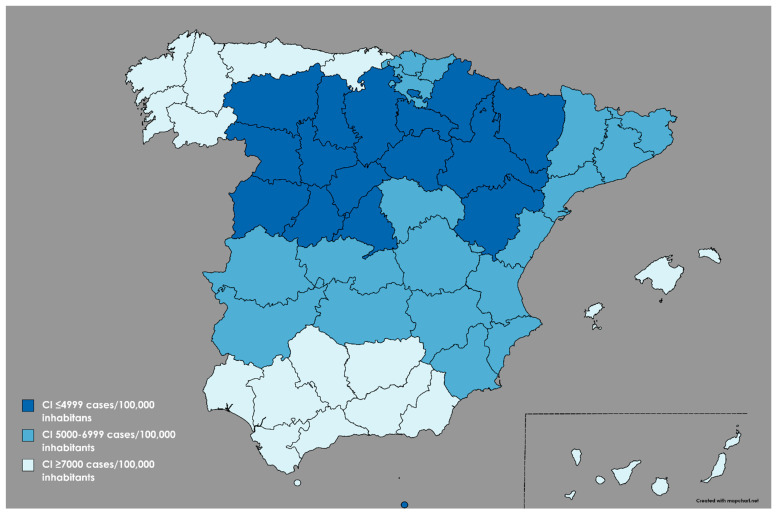
Distribution of the CI regions of COVID-19 cases per 100,000 inhabitants.

**Table 1 jcm-11-02227-t001:** Descriptive characteristics of the sample based on the CI of COVID-19 cases from the different regions.

	Regions		
	Low CI	Medium CI	High CI
**Sex**			
Male	250 (14.62)	330 (19.30)	265 (15.50)
Female	261 (15.26)	315 (18.42)	289 (16.90)
**Age (years)**	43.57 ± 9.71	42.76 ± 10.42	44.42 ± 9.50
**Professional category**			
Physician	151 (8.83)	183 (10.70)	140 (8.19)
Nurse	152 (8.89)	157 (9.18)	144 (8.42)
EMT	204 (11.93)	303 (17.72)	258 (15.09)
Others	3 (0.18)	4 (0.23)	11 (0.64)
**EMS work experience (years)**	15.00 ± 9.09	14.89 ± 9.45	15.80 ± 8.83
**Change of working conditions**			
Yes	286 (16.72)	344 (20.12)	290 (16.96)
No	224 (13.10)	302 (17.66)	264 (15.44)
**Previous diagnosis of COVID-19**			
Yes	442 (25.85)	468 (27.37)	407 (23.80)
No	85 (4.97)	147 (8.60)	161 (9.41)
**Living with minors/dependents**			
Yes	270 (15.79)	363 (21.23)	283 (16.55)
No	234 (13.68)	274 (16.02)	286 (16.73)
**Stress**	21.10 ±10.94	20.86 ± 10.76	19.88 ± 11.56
**Anxiety**	13.34 ± 11.08	13.16 ± 11.00	12.74 ± 11.46
**Depression**	16.19 ± 10.50	15.77 ± 11.17	15.31 ± 11.58
**Self-efficacy**	71.24 ± 15.31	70.42 ± 15.34	70.81 ± 16.62

Values are expressed as mean ± standard deviation or frequencies (percentages). Abbreviations: CI—Cumulative Incidence; EMT—Emergency Medical Technicians; EMS—Emergency Medical Service; COVID-19—Coronavirus Disease-19.

**Table 2 jcm-11-02227-t002:** Level of stress, anxiety, depression and self-efficacy according to sex and the CI regions of COVID-19 cases.

Regions	Sex		*p*-Value (Sex × Region)
	Male	Female	
**Stress**			
Low CI	20.23 ± 11.17 ^&,a^	20.01 ± 10.61	0.135
Medium CI	18.92 ± 10.49 ***^,&^	22.87 ± 10.70 ***	
High CI	17.71 ± 11.67 **^,&,a^	21.94 ± 11.05 **	
**Anxiety**			
Low CI	12.36 ± 11.22 *	14.33 ± 10.87 *	0.442
Medium CI	11.76 ± 10.45 **	14.57 ± 11.35 **	
High CI	10.83 ± 11.13 **	14.54 ± 11.49 **	
**Depression**			
Low CI	15.66 ± 10.68	16.75 ± 10.28	0.155
Medium CI	14.69 ± 10.62 **	16.85 ± 11.62 **	
High CI	13.41 ± 11.11 ***	17.10 ± 11.72 ***	
**Self-efficacy**			
Low CI	73.34 ± 14.60 **	69.46 ± 15.32 **	0.282
Medium CI	71.32 ± 15.28	69.54 ± 15.37	
High CI	71.23 ± 16.66	70.33 ± 16.55	

Values are expressed as mean ± standard deviation. Abbreviation: CI—Cumulative Incidence. * *p* < 0.05 between sexes in the same CI region. ** *p* < 0.01 between sexes in the same CI region. *** *p* < 0.001 between sexes in the same CI region. ^&^
*p* < 0.05 between CI regions in the same sex. ^a^
*p* < 0.05 in the post-hoc analysis (Bonferroni test).

**Table 3 jcm-11-02227-t003:** Level of stress, anxiety, depression and self-efficacy according to professional categories and the CI regions of COVID-19 cases.

Regions	Professional Categories				*p*-Value (Category × Region)
	Physician	Nurse	EMT	Other	
**Stress**					
Low CI	18.36 ± 10.70 **^,a^	20.12 ± 10.70 **	23.08 ± 11.30 **^,a^	16.67 ± 8.33 **	0.413
Medium CI	19.82 ± 10.80	21.08 ± 10.64	21.36 ± 10.84	21.00 ± 6.00	
High CI	18.77 ± 11.41 **	18.11 ± 11.76 **^,b^	21.64 ± 11.27 **^,b^	16.18 ± 12.79 **	
**Anxiety**					
Low CI	11.89 ± 10.28 **^,a^	10.66 ± 10.25 **^,b^	16.35 ± 11.50 **^,a,b^	8.00 ± 5.29 **	0.701
Medium CI	11.96 ± 10.7 *	11.66 ± 10.65 *^,b^	14.65 ± 11.20 *^,b^	13.50 ± 9.00 *	
High CI	10.96 ± 11.18 ***^,a^	10.49 ± 11.27 ***^,b^	15.00 ± 11.30 ***^,a,b^	11.64 ± 12.80 ***	
**Depression**					
Low CI	14.76 ± 11.19 **^,a^	15.07 ± 10.50 **	18.12 ± 10.42 **^,a^	6.67 ± 1.15 **	0.504
Medium CI	14.89 ± 11.89	14.57 ± 10.87	16.90 ± 11.12	17.00 ± 8.41	
High CI	13.97 ± 11.07 **^,a^	13.64 ± 12.05 **^,b^	17.18 ± 11.44 **^,a,b^	10.18 ± 8.65 **	
**Self-efficacy**					
Low CI	70.17 ± 16.19	72.48 ± 14.19	71.83 ± 15.43	78.00 ± 7.55	0.192
Medium CI	70.31 ± 14.70	70.43 ± 15.96	70.59 ± 15.36	59.50 ± 20.29	
High CI	70.80 ± 17.42	72.80 ± 16.18	69.45 ± 16.53	77.00 ± 11.17	

Values are expressed as mean ± standard deviation. Abbreviation: CI—Cumulative Incidence; EMT—Emergency Medical Technicians. * *p* < 0.05 between professional categories in the same CI region. ** *p* < 0.01 between professional categories in the same CI region. *** *p* < 0.001 between professional categories in the same CI region. ^a,b^
*p* < 0.05 in the post-hoc analysis (Bonferroni test).

**Table 4 jcm-11-02227-t004:** Level of stress, anxiety, depression and self-efficacy according to change in working conditions and the CI regions of COVID-19 cases.

Regions	Change of Working Conditions		*p*-Value (Change × Region)
	Yes	No	
**Stress**			
Low CI	22.32 ± 10.68 *	19.74 ± 11.14 *	0.359
Medium CI	22.55 ± 10.42 ***	18.91 ± 10.85 ***	
High CI	21.98 ± 11.60 ***	17.60 ± 11.09 ***	
**Anxiety**			
Low CI	14.34 ± 10.94 *	12.15 ± 11.15 *	0.408
Medium CI	14.95 ± 11.28 ***	11.14 ± 10.37 ***	
High CI	14.50 ± 11.92 ***	10.78 ± 10.65 ***	
**Depression**			
Low CI	17.40 ± 10.63 **	14.69 ± 10.16 **	0.501
Medium CI	16.97 ± 11.29 **	14.41 ± 11.94 **	
High CI	17.18 ± 11.85 ***	13.22 ± 10.94 ***	
**Self-efficacy**			
Low CI	70.77 ± 15.65	71.80 ± 14.93	0.876
Medium CI	69.56 ± 15.51	71.36 ± 15.16	
High CI	69.89 ± 16.69	71.87 ± 16.54	

Values are expressed as mean ± standard deviation. Abbreviation: CI—Cumulative Incidence. * *p* <0.05 between professional categories in the same CI region. ** *p* <0.01 between professional categories in the same CI region. *** *p* <0.001 between professional categories in the same CI region.

**Table 5 jcm-11-02227-t005:** Level of stress, anxiety, depression and self-efficacy according to the previous diagnosis of COVID-19 and the CI regions of COVID-19 cases.

Regions	Previous Diagnosis of COVID-19		*p*-Value (Diagnosis × Region)
	Yes	No	
**Stress**			
Low CI	20.95 ± 10.77	22.72 ± 11.33	0.695
Medium CI	20.18 ± 10.76	22.03 ± 9.74	
High CI	19.39 ± 11.72	20.56 ± 11.07	
**Anxiety**			
Low CI	13.10 ± 10.84	15.61 ± 11.77	0.790
Medium CI	12.41 ± 10.86	14.07 ± 10.49	
High CI	12.32 ± 11.54	13.53 ± 11.57	
**Depression**			
Low CI	16.25 ± 10.51	16.69 ± 9.95	0.740
Medium CI	15.12 ± 11.08	16.98 ± 10.91	
High CI	14.87 ± 11.71	15.94 ± 11.44	
**Self-efficacy**			
Low CI	70.59 ± 14.82	74.48 ± 15.37	0.329
Medium CI	69.87 ± 15.30	70.72 ± 15.38	
High CI	70.83 ± 17.15	70.80 ± 14.96	

Values are expressed as mean ± standard deviation. Abbreviation: COVID-19—Coronavirus Disease-19; CI—Cumulative Incidence.

**Table 6 jcm-11-02227-t006:** Level of stress, anxiety, depression and self-efficacy according to living with minors/dependents and the CI regions of COVID-19 cases.

Regions	Living with Minors/Dependents		*p*-Value (Minor/Dependent × Region)
	Yes	No	
**Stress**			
Low CI	20.37 ± 10.60	22.52 ± 11.26	0.674
Medium CI	20.14 ± 10.78	21.78 ± 10.56	
High CI	19.45 ± 11.33	20.18 ± 11.68	
**Anxiety**			
Low CI	12.43 ± 10.39 *	14.77 ± 11.64 *	0.260
Medium CI	12.57 ± 10.97	13.94 ± 11.03	
High CI	12.63 ± 10.99	11.69 ± 11.82	
**Depression**			
Low CI	16.33 ± 10.31	16.21 ± 10.61	0.959
Medium CI	15.91 ± 11.53	15.62 ± 10.91	
High CI	15.51 ± 11.49	14.99 ± 11.63	
**Self-efficacy**			
Low CI	71.14 ± 15.15	70.78 ± 15.84	0.520
Medium CI	70.01 ± 15.42	70.94 ± 15.30	
High CI	69.96 ± 16.95	71.86 ± 15.99	

Values are expressed as mean ± standard deviation. Abbreviation: CI—Cumulative Incidence. * *p* < 0.05 between professional categories in the same CI region.

**Table 7 jcm-11-02227-t007:** Level of stress, anxiety, depression and self-efficacy according to age, EMS work experience and the CI regions of COVID-19 cases.

	Regions	Stress	Anxiety	Depression	Self-Efficacy
**Age**	Low CI	−0.109 *	−0.104 *	−0.097 *	0.001
	Medium CI	−0.140 ***	−0.176 ***	−0.183 ***	−0.045
	High CI	−0.087 *	−0.101 *	−0.109 **	−0.028
**EMS work experience**	Low CI	−0.099 *	−0.122 **	−0.094 *	0.085
	Medium CI	−0.144 ***	−0.150 ***	0.206 ***	0.002
	High CI	−0.087 *	−0.133 **	−0.092 *	−0.008

Values are expressed as Spearman’s r. Abbreviation: IA—Cumulative Incidence; EMS: Emergency Medical Service; * *p* < 0.05. ** *p* < 0.01. *** *p* < 0.001.

## Data Availability

Data for this study are available by contacting the corresponding authors.

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
