# Peer review of "Influence of the Cumulative Incidence of COVID-19 Cases on the Mental Health of the Spanish Out-of-Hospital Professionals"

_jcm, 2022, doi:10.3390/jcm11082227_

Round 1
Reviewer 1 Report
Dear editors and authors,
The aim of this study was to assess levels of stress, anxiety and depression in health professionals of Spanish Emergency Medical Services and compare the finding to the cumulative incidence of COVID-19 in the regions in which they worked. The authors have enrolled a high number of participants which is one of the strengths of their study.
Study methods seem appropriate, as well as ethic statements regarding them. Scales used in the study are validated and also appropriate to the study objectives.
Cited references are current and do not include an abnormal number of self-citations.
I find this manuscript interesting and informative. However, I have some comments and suggestions which could further improve the text.
- There are many abbreviations, some of which were not explained (SEM and PS).
- The introduction part should have a section on the mental health of the health professionals as this profession usually has high stress levels. This should also be addressed in the discussion.
- In the Results part of the manuscript, there is an overlap between data presented in text and tables which should be avoided.
- At the start of the Discussion part, the authors' wording suggest that they have proved causality, which was not possible with this study as they write in limitations of the study, too.
- The authors should compare their results with similar studies or with the Spanish general population.
Reviewer 2 Report
Overall manuscript is looking good. Few suggestions:
Line 26: Please explain what SEM stands for
Line 40 Keywords: Efficacy is spelled wrong
Line 62: What PS ( ?Psychological stress) stands for
